# Effects of *Haematococcus pluvialis* Addition on the Sensory Properties of Plant-Based Meat Analogues

**DOI:** 10.3390/foods12183435

**Published:** 2023-09-15

**Authors:** Meng Liu, Yanli Wang, Laijing Zhu, Xiangzhong Zhao

**Affiliations:** School of Food Science and Engineering, Qilu University of Technology (Shandong Academy of Sciences), Jinan 250353, China; 15864530661@163.com (M.L.); 15106614358@163.com (Y.W.); 15725152458@163.com (L.Z.)

**Keywords:** *Haematococcus pluvialis*, plant-based meat analogues, electronic nose, different color value

## Abstract

Due to the increase in population and the deficiency of land resources, the cost of raising livestock is gradually increasing. Plant-based meat analogues (PBMAs) are considered excellent substitutes for animal meat. Our research investigated the effect of *Haematococcus pluvialis* (HP) on gluten-based soybean and wheat PBMA with contents of 1%, 3%, 5%, and 7%. Compared with the control group, HP significantly improved the color of the extrudates, showed visual characteristics similar to red meat, and achieved a soft texture and apparent rheological properties. The 7% HP had negative effects on the organizational degree and viscosity. In addition, the E-nose indicated that the different contents of HP changed the flavor of the extrudates. The extrudates with 3% and 5% HP were most similar to each other among all of the extrudates for volatile compounds, and the extrudates with 1% HP and 7% HP had significantly different flavors compared to the control group. Furthermore, 20 different volatile compounds were compared according to their retention indices and retention areas. The results showed that the proportions of alcohol, ester, terpenes, acid, and furan were increased. When the threshold was referenced, HP was considered to provide PBMAs with grassy and healing grain flavor properties. Therefore, the results proved that the addition of HP can improve PBMAs sensory properties.

## 1. Introduction

Plant-based meat analogues (PBMAs) are popular as main new materials in the context of dwindling food resources [1]. As a source of raw material for PBMAs, plants require lower land and feed costs than traditional animal resources, and the concerns about environmental pollution and the greenhouse effect are reduced. In addition, plant protein has a healthier ingredient formula and excellent amino acid composition compared to animal protein, and antibiotic residues are avoided. Therefore, the market of meat analogues is developing rapidly, and the outlook is excellent.

PBMAs are mainly produced by twin-screw extruders. By double-screw stirring and reflection using high temperatures, its secondary structure is exposed. Then, passing through the cold region, the evenly mixed proteins are crosslinked, again contributing to fibering [2]. Mimicking the fiber characteristics of animal meat is the key to developing meat analogues, and improving the organizational degree is a research hotspot. Meanwhile, perfect sensory properties are also essential for selective purchasing, such as color, aroma, and taste. These are also key factors in a consumer’s choice of product [3]. Furthermore, the visual characteristics receive attention as the first key aspect of selection. Currently, natural pigments, for instance, red koji and beetroot, are mainly used as colorants of PBMAs [4]. Leghemoglobin is also popular due to its ability to change the color of meat before and after cooking. Many companies, such as Impossible Foods, have already chosen to use leghemoglobin in their products [5]. At present, the production of leghemoglobin mainly depends on microbial fermentation [6]. The productional process requires techniques to improve its efficiency. In addition to the use of pigments, small-sized microalgae with simple cultivation conditions are used. These may have the potential to improve the color properties of PBMAs. The nutrients in microalgae also need to be determined. 

Microalgae are microorganisms that can grow through photosynthesis and have a positive effect on the greenhouse effect; they have attracted much attention in recent years [7]. The nutritional benefits of microalgae biomass are generally popular due to their nutritional compounds [8] and perfect functional features [9], such as single-cell proteins (*Arthospira platens* and *Chlorella vulgaris*) [10], β-carotene (*Dunaliella*) [11], astaxanthin (*Haematococcus pluvialis*) [12], and docosahexaenoic acid (*Schizocohytrium* sp.) [13]. Meanwhile, microalgae residues are usually discarded, resulting in a waste of resources. Thus, directly adding microalgae is considered to be a new production method [14,15,16]. *Haematococcus pluvialis* (HP) is rich in astaxanthin, which shows as a dark red color with a slight smell of seaweed [17]. It is a potential source for the production of astaxanthin, which is known as Super Vitamin E and has strong antioxidant activity [18]. However, astaxanthin requires harsh storage conditions and complex extraction steps [19], which restricts its use. The direct use of HP in food production may be one of the strategies to save costs and make efficient use of astaxanthin [20]. In some recent research, it was indicated that microalgae can change the sensory properties of PBMAs [21]. Barkallah et al. [22] used about 1% spirulina in their fish burgers, and the color, smell, and taste of the veggie burgers were improved. Crahl et al. [23] applied spirulina to PMBAs and obtained a dark appearance. Compared with the green microalgae, the HP had a distinct advantage in terms of imitating the color of animal meat.

In this study, unbroken HP was added to PMBAs, aiming to change the color to meet the needs of the fleshy surfaces of the extrudates, and to affect the fiber structure so that intact algal cells were not destroyed by the squeezing process and, thus, could be embedded into the void. Considering that excessive microalgae has a certain influence on the flavor of PMBAs, such as an algal smell and grassy aroma, in addition to using Fourier infrared spectroscopy and a rheometer to characterize their structure and functional properties, the electronic nose was also used to evaluate the influence of HP on the flavor of PMBAs. The optimal amount of algae powder was determined, and the effect of HP on the structure and sensory aspects of PBMAs was analyzed.

## 2. Materials and Methods

HP (25.6% protein, 7.05% ash, 1.96% water, 3.09% astaxanthin) was provided by Yunnan Alkang Co., Ltd. (Chuxiong, China). The HP was stored in refrigerators at −4 °C and vacuumed. SPI was produced by Anyang Tianxiangrui food science and technology Co., Ltd. (Anyang, China). WI was produced using double-screw extrusion equipment produced by Shandong Arrow Machinery Co., Ltd. (Jinan, China), TA. An XTC-20 texture analyzer was produced by Shanghai Bosin Industrial Development Co., Ltd. (Shanghai, China), and an RH-20 rheometer analyzer was produced by Shanghai Bosin Industrial Development Co., Ltd. (Shanghai, China). A Heracles neo-E-nose was produced by France Alpha MOS Co., Ltd. (Toulouse, France); an NR20XE-1601739 color difference meter was produced by Shenzhen Threenh Technology Co., Ltd. (Shenzhen, China); and an EZ-TEST biomechanical tester was produced by Shimadzu Corporation (Kyoto, Japan).

### 2.1. The Preparation of High-Moisture Extrusion

The high-moisture extrusion experiment was conducted using a twin-screw extruder [24] (Shandong Arrow Machinery Co., Ltd.). As shown in Figure 1, after soy protein isolate (SPI), gluten (WI), and complex phosphate were mixed in a mixer at 10 min, HP was added in an amount of 1%, 3%, 5%, and 7%, respectively. The experimental conditions of high water extrusion were set according to reference [25] and slightly adjusted. The extrusion barrel temperatures of the eight zones were set as 40 °C, 60 °C, 80 °C, 120 °C, 140 °C, 160 °C, 150 °C, and 40 °C, as shown in Figure 1. The power of the feeder was set at 12.70 Hz, the rotation power of the screw was set to 30.00 Hz, and the speed of the screw was set to 840 RPM. Water was added to make sure that the moisture content of PBMA was 60%, and the protein content was measured by the Kjeldahl nitrogen determination method [26]. The products were collected and cut in sections, then sealed by vacuum after cooling them to room temperature. Then, extrudates were stored at 4 °C before the test.

### 2.2. Tensile and Water Absorption Analysis

PBMA was placed at room temperature and cut into evenly sized (2 cm × 2 cm × 1 cm) pieces. Wobb’s cutter and an EZ-TEST biomechanical tester (Shimadzu Corporation) were used to determine the maximum shear force in the cross-section and longitudinal directions. The test speed was set as 50 mm/min, and the shear displaced 40 mm. The maximum shear force was measured three times in the parallel experiment for each sample. The result of the organizational degree was determined according to the computational formula found in [27]. The TPA test was performed by using a texture analyzer (Shanghai Bosin Industrial Development Co., Ltd.). The TA/20 cylindrical probe was selected at a specific test speed, and a shape variable was set at 50%. Then, precipitation was retained after centrifuging and weighed to calculate the water absorption of the sample. All experiments were repeated three times.
(1)$$\begin{array}{c} L=\frac{F_{L}}{F_{T}} \end{array}$$

L is the organizational degree, F_L_ is the longitudinal tensile force, and F_T_ is the transverse tensile force.

### 2.3. E-Nose Analysis

PBMA was divided into samples of similar mass (0.5 cm × 0.5 cm × 0.5 cm) for the electronic nose (France Alpha MOS Co., Ltd.), which was weighed with the same quality sample in the headspace bottle and shaken for 20 min at 90 °C. Then, the attempt speed was set at 125 μL/s to sample 45 s at 200 °C. Next, the collecting trap was heated from 40 °C to 240 °C, and the diversion speed was set to 10 mL/min and kept at 50 s. Two-color columns (MXT-5 and MXT-1701) used 150 °C and 250 °C to collect volatile compounds at 340 s. Calibration was performed using a standard solution of n-alkanes (nC6–nC16). The results were presented using principal component analysis and radar mapping. All experiments were repeated three times and measured on the same day.

### 2.4. Electron Microscope Analysis

PBMAs were selected for the parallel extrusion directly on the test surface [28], then cut into 12 mm × 7 mm × 3 mm cubes. After the samples were freeze-dried, gold spray was used for 45 s at 10 mA using a Quorum sputtering coater, and a scanning electron microscope (SEM) (Carl Zeiss AG) was used to scan electron microscopy. The formation characteristics of the fiber structure of the plant meat parallel to the extrusion direction were observed at three magnifications.

### 2.5. Rheological Analysis

The freeze-dried powder, as the experimental material for analysis, was passed through a 100-mesh sieve using a rheometer analyzer (Shanghai Bosin Industrial Development Co., Ltd.); then, the powder was mixed with distilled water and refrigerated overnight at 4 °C [29]. A parallel plate with a diameter of 25 mm was used; the shear rate was 1 mm/s~100 mm/s. The treated adhesive was evenly spread over the gap between the bottom surface of the rheometer and the parallel plate to test the sample’s viscosity. The storage modulus (G′) and loss modulus (G″) were measured by using a rheometer analyzer (Anton Paar Co., Ltd., Graz, Austria). The angular frequency was set at 0.1–100 rad/s. All the experiments were repeated three times.

### 2.6. Color Difference Analysis

The color of PBMA was determined using a color difference meter (Shenzhen Threenh Technology Co., Ltd.) in triplicate. The values of *L** (brightness), *a** (redness), *b** (yellowness), and ΔE were measured [30]. Before the sample was tested, the machine was corrected using a white standard. All of the experiments were repeated three times.
(2)$$\begin{array}{c} ∆E={\left[{\left({∆L}^{*}\right)}^{2}+{\left({∆a}^{*}\right)}^{2}+{\left({∆b}^{*}\right)}^{2}\right]}^{\frac{1}{2}} \end{array}$$

### 2.7. Sensory Evaluation

Sensory evaluation of the color and odor was conducted by 7 professionals in the meat analogues field (A group) and 7 people with no experience in this field (E group). The panelists had experience in sensory evaluation, and their average age was 31. All of the tested samples were randomly placed to avoid subjective thoughts interfering with the sensory evaluation result. The evaluate criteria are shown in Appendix A. On the other hand, the samples used for odor evaluation needed to be cut and heated.

### 2.8. Statistical Analysis

SPSS26 (IBM Corporation, Armonk, NY, USA) was used for statistical analysis with a limit of statistical significance of *p* < 0.05. All data were plotted using Origin 2021 software.

## 3. Results

### 3.1. Appearance and Color

The content of the HP powder showed that it was mixed well, with the appropriate proportions of soy protein isolate, gluten, and complex phosphate. After being poured into a twin-screw extruder, HP was crowded into fiber in the extrudates, as shown in Figure 1. The appearance and color of the PBMAs with HP were directly affected. Figure 2 shows that the PBMA originally changed to red, like beef, with the addition of the HP from 1% to 7%. But with the addition of more than 5% HP, the color was too dark to provide a satisfactory visual experience. On the other hand, the amount of fiber was affected negatively. The sample with 7% HP added made the fibrous structure compact, and the elasticity of the fiber was improved when the amount of fiber was decreased sharply. The result was similar to that obtained in the research of Caporgno [31]. The intact cell wall of the HP after extrusion was arranged in the fiber of the PBMA; it could be seen through an optical microscope (400×) that this arrangement loosened the fibrous structure of PBMA.

### 3.2. Texture Analysis

It was found that there was an influence of the amount of HP on the hardness and elasticity indices after the TPA test. As shown in Table 1, indicating that the addition of algal meal had affected the texture properties of PBMA, the hardness at 7% HP decreased by 30.13% (*p* < 0.05), the chewiness at 7% HP decreased by 56.49% (*p* < 0.05), and the cohesiveness at 7% HP decreased by 46.59% (*p* < 0.05). With the addition of HP, the hardness of PBMA decreased gradually, which is a similar result to that of the research conducted by Crahl [23]. The reason may be that the embedding of HP caused holes to appear in the PBMAs, contributing to their soft characteristics. In addition, the elasticity of PBMA showed a slightly increased trend with the addition of HP, which may be attributed to the hard cell walls which could not be broken. Thus, the PBMAs with HP added were rich in resilience after deformation. 

Judging by the content of transverse tensile forces and longitudinal tensile forces, the degree of the microstructure showed an opposite trend to the increase in HP, and the organizational degrees of 3% and 5% algae powder were similar. The organizational degree increased by 10.43% (*p* < 0.05) at 5% HP. Corresponding to the apparent features were the organizational degree, which decreased to 1.42 when the additional amount of HP was 7%, a decrease of 12.88% (*p* < 0.05). This indicated that the addition of HP (more than 5%) might have reduced the fiber of PBMA and had an effect on the crosslinking reaction between proteins.

### 3.3. The Content and Proportion of Volatile Compounds Analysis

This study used Heracles to explore the influence of HP on the meat flavor of the PBMAs, mainly to identify the different compounds and the trend of change [32]. As shown in Figure 3, the contribution rates of the first principal component (P1) and the second principal component (P2) reached 98.819% according to the principal component analysis (PCA) diagram, which could represent the sample information more completely. The Mahalanobis distance in the figure reflected that the additional amount of 1% HP powder was farthest from that of the control group, so the smell difference was the largest. The odors of samples with 3% and 5% added HP had compact positions; therefore, the smells were similar. Moreover, the arrow in Figure 3 reflects the contents of the differential compounds in PBMA.

Heracles uses a double-hydrogen flame ionization detector, which belongs to a mass flow rate sensitive detector. Thus, the content contrasts of the differential compounds were determined by the retention parameters. Appendix A showed the gas chromatograms of these samples. As shown in Figure 4a–c, a total of twenty important different volatile compounds were detected by the Arochembase data library, including nine alcohol, four ketones, two esters, two acids, two terpenes, and three other compounds, and changes were identified after the addition of HP.

Among these, alcos are one of the main volatile components of microalgae [33]. The addition of HP increased the variety of alcohol in PBMA, as shown in Figure 4d. 2-amul alcohol (green grass) is produced by fermentation; the amount of algae needs to rely on alcoholic fermentation to produce energy, and has grassy fragrance characteristics. Therefore, 2-amul alcohol, as a landmark substance, showed a step-type increase. Associated with the former generation pathway is ethanol, which, as an end product of the fermentation process, was increased with the addition of HP. 2-hexol (alcohol), which was the most prevalent volatile compound, underwent a downward process with the addition of HP. This was a compound with flavors of cauliflower and wine, and was mainly used to configure a berry flavor. The reason might be because the HP increased the gaps in the PBMAs, and the structures lost their tightness, thus reducing the retention rate of the volatile compounds [34]. It is worth noting that the content of acetic acid (vinegar), as the product of biological fermentation, increased with the addition of algal powder, which may be due to the fact that, during the storage of PBMA, HP developed its ethanol content through biological fermentation. Moreover, it is worth paying attention to the fact that the addition of 7% HP promoted the retention of 2-acetyl -1-pyrroline (2-AP) (nut), which increased the content of this compound compared with the control group. This compound, mainly rooted in grain, has a rice fragrance and a healing odor of ham and nut [35]. It shows a particularly low sensory threshold. It may be that the compounds in PBMA including WI were characteristic of their aromas. In addition, the substance is detected in beef and lamb [36,37], and may be a component of main flavor compounds. By contrast to the control and 1% HP samples, it was noticed that a small amount of HP could reduce the contents of volatile compounds in PBMAs, causing a loss of flavor and making the undesirable flavors of the food change more quickly than if the flavors were reversed by adding the HP powder.

But the olfactory characteristics of PBMA could not be assessed by content alone. The relative content of volatile substances was also essential for the analysis. The relative reserved area is shown in Table 2. The contributions of differential compounds to the flavor of plant-based meat were roughly judged by comparing the relative contents. The addition of HP linearly reduced the proportion of 2-hexol (cauliflower), linalool (foeniculum vuglare), α-terpineol (orange), damascus ketone (fruits), and 2-geng ketone (agaric), indicating that HP might reduce the abilities of the five scented substances to bind to protein after extrusion. 2-geng ketone and 2-hexol make up the volatile compounds of fishy substances; these were reduced, helping to improve the flavor of PBMA. Additionally, cis-3-hexen-1-ol (leaf), n-hexyl alcohol (alcohol), ethanol (alcohol), anti-2-pentol (agaric), 2-amyl alcohol, isoamyl acetate (scallion), ethyl crotonate (fruits), acetic (vinegar), and 2-ethyl furan (scorch) were increased. These different odor compounds imparted complex odor properties to PBMAs with HP, such as a grassy fragrance and a healing scent similar to grain. However, the masking effect on the small bean substance will require further study.

### 3.4. Microstructure

The microstructure of the PBMA with algal powder added was compared with the SEM. This comparison, shown in Figure 5, reflects the complete HP being protected from extrusion due to its stiff cell wall; its wrapping side by side in the fiber contributed to some gaps contracting, leading to extrusion without HP. Moreover, the flexibility of the extrudates was improved, maybe due to the fact that HP has a hard wall that can be considered to be a support rod. During heating, the chemical bonds inside the protein were broken, so the chains were unfolded. At this time, the double screws furthered the mixing of the fused protein with HP powder. When the material reached the cooling molding area, giant algal cells prevented the normal crosslinking of the protein, and the chemical bonds were broken so that the fiber structure and amount were changed. As shown in Figure 5, a small number of globular HP and pore networks were observed among the longitudinal section, and the changes in the structure and hardness decreased gradually. When the contents were 1% and 3%, the microstructure of the extrudate presented more compact and larger pore networks, which may have improved the soft characteristics. But when the ratio was 5%, the gaps became smaller and more compact, which might have resulted in the fiber improving slightly, as shown in Table 1. On the other hand, a dense and small porous structure similar to sponge had a positive effect on the improvement of the hydraulic and oil-holding power. This can be considered to be a functional characteristic, providing PBMAs with an advantage in future research.

### 3.5. Rheological Analysis

In static rheological experiments, non-Newtonian fluids exhibit pseudoplastic flow characteristics. The main manifestation is the shear-thinning phenomenon due to the polymerization between unraveled macromolecules. The difference in HP content and its effect on viscosity (*p* < 0.05) of PBMA is shown in Figure 6; when the shear rate increased, the viscosity decreased substantially. Viscosity can be considered to be the internal friction force during fluid motion. HP is globular and large in size, which caused PBMAs to take on a puff-pores network instead of a compact crosslinked one. As shown in Figure 5, the connections between the proteins became loosely prone to disruption; therefore, the friction of the sample decreased and the viscosity was affected negatively. It was discovered that 3% HP can impart a greater level of viscosity to PBMA after stabilization, in contrast to other samples. On the other hand, this result may indicate the stability of the protein structure. Shear cannot cause the protein structure to be completely destroyed, and the chemical bonding force between proteins was different when HP was added.

On the other hand, the rheological experiment result showed a change in the storage modulus (G′) and the loss modulus (G″). G′ represents the sample’s elasticity. Energy is temporarily stored, then can be used. As shown in Figure 6b, the addition of HP firstly decreased G′, but then caused a gradual improvement. In addition, 3% HP added to the extrudate showed the highest value compared with the others, proving that 3% HP can lead to benefits in the formation of the gel structure. But all extrudates with added HP showed lower values than the control group. The reason might be that the porous structure causes the insides of extrudates to loosen; this guess is consistent with the SEM analysis. The G″ represents the sample’s viscosity, and energy could not be used. The result showed a similar trend to G′ in Figure 6c, and the extrudate with 3% of added HP showed a higher value than the control group, so the extrudate had better viscosity. Furthermore, all G′ samples were larger than G′, indicating that all samples exhibited solid properties and mainly contained elastic components. The change in G’ was more obvious than that in G″, proving that the addition of HP had a significant effect on the elasticity, compared to the viscosity, of PBMAs.

### 3.6. Analysis of Color Difference Value

The color of PMBAs depends on the raw material and the Maillard reaction during extrusion. As shown in Table 3, which shows the state of the dry powder before extrusion, with a brightness value (*L**) indicating a significant difference between the sample and whiteboard, the addition of HP obviously reduced the *L** of raw powder. The yellow value *a** of the sample increased with the content of HP powder, indicating that the addition of HP caused the color of the powder before extrusion to appear red. Compared to cooked beef, the *L** of extrudates with HP was obviously reduced. However, the *a** of extrudates with HP increased near cooked beef. These color difference values indicated that the visual characteristics of the PBMA with the addition of 5% HP were nearer to those of raw beef. To achieve contrast to the commonly used pigment in foods with red color, monascus red, for example, needs to be used. Therefore, the different coloring effects are shown in Table 3. The extrudates with monascus red added indicated a nearer *L** to thoroughly cooked beef compared to HP, but the ability to simulate *a** was not outstanding. In the dry powder state, the redness raw material mixed with the HP powder was not significantly affected.

### 3.7. Sensory Evaluation

The sensory evaluation results are shown in Table 4. The color score showed a gradual upward trend, and the highest value appeared when 1% and 3% HP were added to the PBMAs. The report indicated that for most of these two types of evaluators, the extrudate with 3% HP added was closer to traditional meat; their similar colors can increase their appeal as substitutes for animal meat. The result was relative to the analysis of color difference. But the evaluators diverged in terms of the selection of the meat samples which they liked least. The E group considered that the extrudates with 7% HP added appeared too dark red, but also had the characteristics of animal meat. On the other hand, the A group gave the lowest scores to extrudates with 7% HP added. They thought that the control group was more able to imitate the color of cooked chicken compared to the extrudates with 7% HP added.

The odor scores were highest when 7% HP was added to the PBMAs. The odor evaluation score indicated that the addition of HP can affect the odor of PBMA. Furthermore, the evaluators believed that 7% HP would lead to a green odor, decreasing the soybean fishy odor; a slight leaf flavor caused the evaluators to designate the samples as more healing and healthier. In addition, the control group gave the lowest scores in the odor evaluation. This was because the evaluators thought that the soybean’s fishy smell would have a negative effect on appetite development.

## 4. Conclusions

This research demonstrates that different contents of HP had significant effects on the sensory properties of SPI-WI-based meat analogues. The most noteworthy result of this report is that the visual characteristics of PBMA with added HP were similar to red animal meat. The hypothesis that HP has an advantage when used for meat color imitation was proven. On the other hand, the extrusion process was unable to destroy intact walls of microalgae, and a denser porous structure was formed in PBMA. The porous structure gave extrudates soft properties. Meanwhile, the addition of HP affected the flavor of PBMA. Firstly, algae itself contains more fishy substances, and the addition of HP caused the contents of alcohol compounds to show obvious changes. The proportions of ester, terpenes, acid, ketone, and furan appeared to float, which may be due to the flavor substances carried by HP or the provision of suitable precursors. Secondly, as shown in this study, the contents and proportions of some volatile compounds were not relative. This result might become the key to evaluating the flavor of PBMAs. A denser porous structure can promote the volatilization of flavor, so the odor threshold needs to be determined. In particular, the addition of HP caused PBMAs to show pseudoplastic flow characteristics, and G′ and G″ were affected. 

This research verified the possible use of HP as a biological compound in extrudates instead of natural pigments. We summarized flesh-sensitive properties such as color, texture, odor, viscosity, and organizational degree. The functions of HP’s bioactive aspects need to be assessed in subsequent research to promote the development of beneficial diets in the future.

## Figures and Tables

**Figure 1 foods-12-03435-f001:**
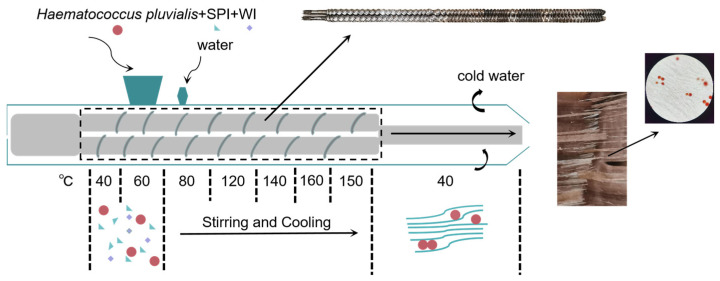
The procedure chart of how plant-based meat analogues (PBMA) were extruded with *Haematococcus pluvialis* (HP) by a twin-screw extruder and the microstructure and appearance of the extrudate.

**Figure 2 foods-12-03435-f002:**
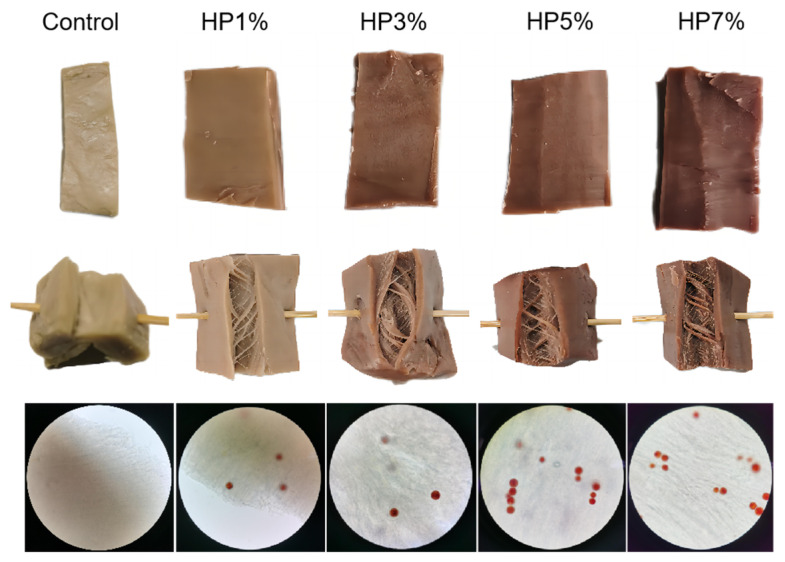
Appearances and colors of the PBMAs with different contents of HP.

**Figure 3 foods-12-03435-f003:**
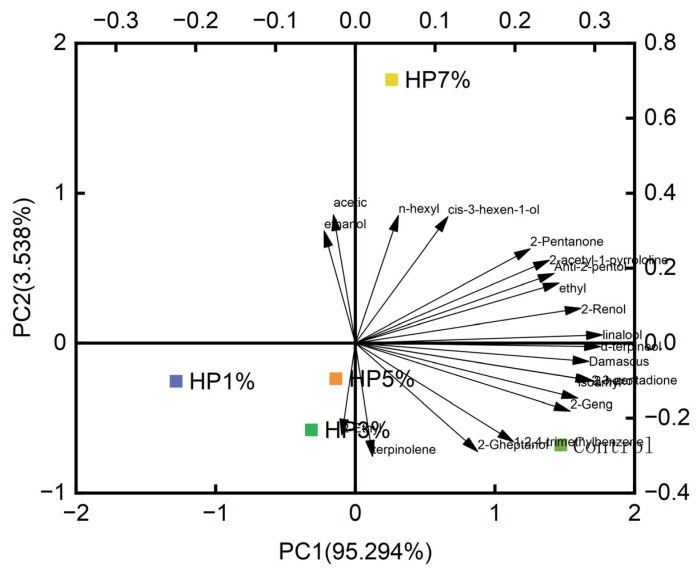
PCA for PBMA with different HP contents and volatilized compounds added.

**Figure 4 foods-12-03435-f004:**
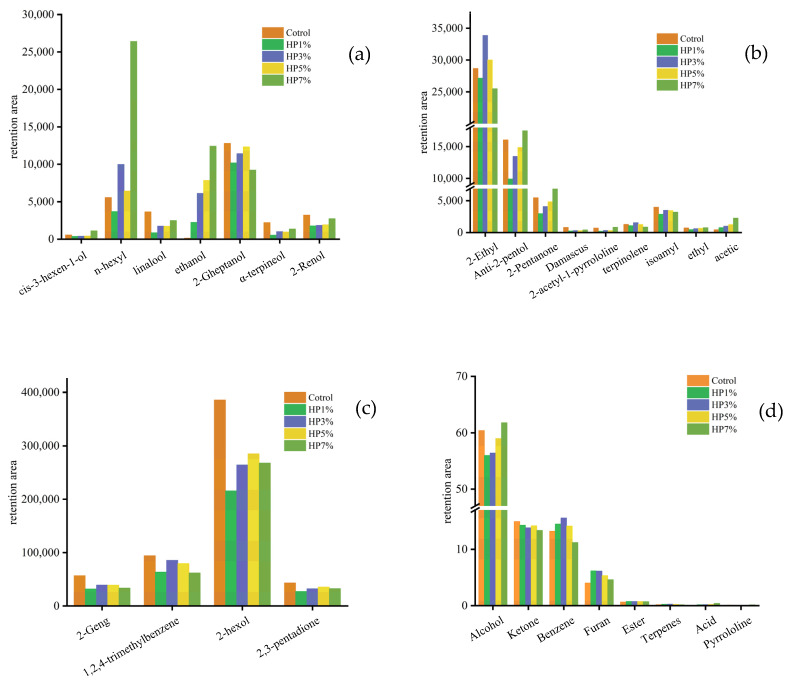
The contents of differential volatile compounds with different concentrations of HP (the content of differential compounds was assessed by retention parameters). (**a**–**c**) represent the retention areas of differential volatile compounds in PBMAs with different added contents of HP; (**d**) represents the retention areas of volatile compounds in PBMAs with different added contents of HP.

**Figure 5 foods-12-03435-f005:**
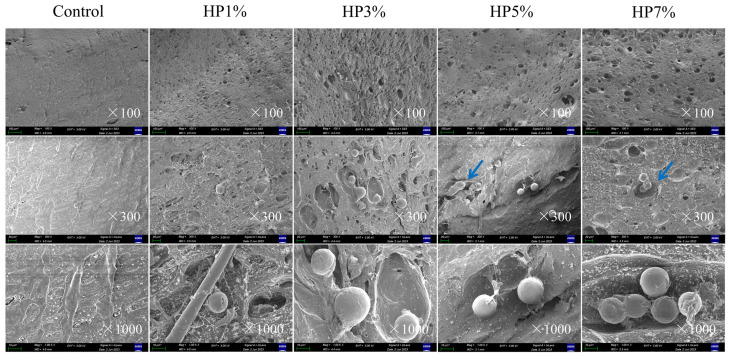
The SEM microstructure of PBMA with different added contents of HP (the side by side HP was pointed by blue arrows).

**Figure 6 foods-12-03435-f006:**
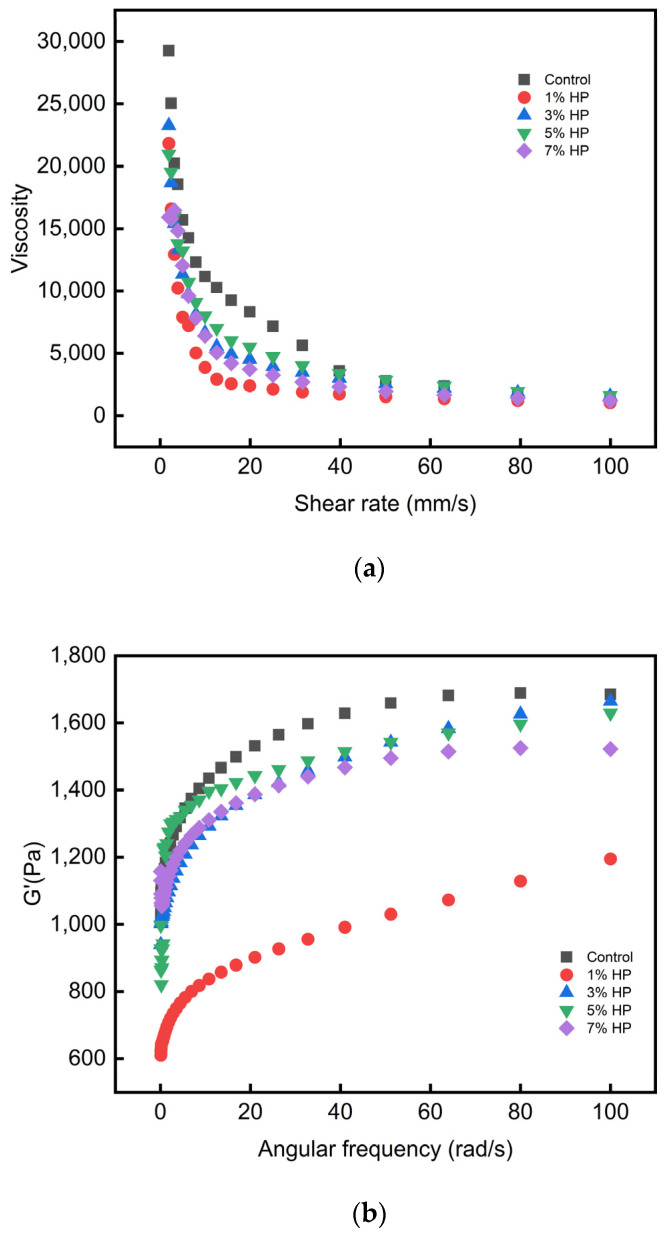
The viscosity (**a**), G′ (**b**), and G″ (**c**) of PBMA with different concentrations of HP.

**Table 1 foods-12-03435-t001:** Effect of HP content on the mass structure and protein content of PBMA.

Mass Structure	HP Content (%)				
	0	1	3	5	7
Hardness (gf)	11,976.62 ± 25.32 ^a^	10,078.48 ± 8.99 ^b^	9952.64 ± 22.30 ^c^	8465.25 ± 15.80 ^d^	8367.95 ± 20.68 ^e^
Flexible	0.92 ± 0.01 ^b^	0.96 ± 0.02 ^a^	0.97 ± 0.08 ^a^	0.97 ± 0.02 ^a^	0.98 ± 0.01 ^a^
Chewiness (gf)	9774.10 ± 104.28 ^a^	8398.90 ± 279.63 ^b^	4254.26 ± 17.17 ^c^	4270.79 ± 20.80 ^c^	4252.36 ± 17.45 ^c^
Cohesiveness	0.88 ± 0.01 ^a^	0.86 ± 0.02 ^a^	0.55 ± 0.06 ^b^	0.46 ± 0.01 ^c^	0.47 ± 0.02 ^c^
Longitudinal tensile force (N)	40.06 ± 2.11 ^b^	45.93 ± 2.01 ^a^	34.82 ± 2.94 ^c^	27.28 ± 2.31 ^d^	34.31 ± 2.70 ^c^
Transverse tensile force (N)	24.72 ± 2.94 ^a,b^	27.98 ± 1.80 ^a^	22.99 ± 1.86 ^b^	15.21 ± 1.54 ^c^	24.25 ± 2.72 ^a,b^
Organizational degree	1.63 ± 0.17 ^a,b^	1.64 ± 0.07 ^a,b^	1.61 ± 0.19 ^a,b^	1.80 ± 0.05 ^a^	1.42 ± 0.10 ^b^
Protein content (%)	33.07 ± 0.46 ^b^	33.59 ± 0.25 ^b^	35.27 ± 1.03 ^a^	33.02 ± 0.61 ^b^	31.32 ± 0.16 ^c^

Note: Different superscript letters in the same columns indicate significant differences (*p* < 0.05).

**Table 2 foods-12-03435-t002:** The proportions of volatile compounds in PBMAs with different added HP contents.

Categories	CAS Number	Volatile Compounds	RI-5	RI-1701	Molecular Formula	HP Content (%)				
						0	1	3	5	7
Alcohol	626-93-7	2-hexol	801	900	C_6_H_14_O	54.20	49.24	48.06	50.72	48.49
	928-96-1	cis-3-hexen-1-ol	861	976	C_6_H_l2_O	0.08	0.08	0.07	0.07	0.20
	111-27-3	n-hexyl alcohol	870	980	C_6_H_14_O	0.78	0.84	1.81	1.15	4.78
	78-70-6	linalool	1099	1095	C_10_H_18_O	0.51	0.20	0.32	0.30	0.45
	64-17-5	ethanol	437	564	C_2_H_5_OH	0.02	0.51	1.11	1.39	2.25
	543-49-7	2-Gheptanol	901	1000	C_7_H_16_O	1.80	2.33	2.08	2.20	1.67
	98-55-5	α-terpineol	1189	1300	C_10_H_18_O	0.31	0.12	0.18	0.17	0.25
	628-99-9	2-Renol	1102	1200	C_9_H_20_O	0.45	0.40	0.33	0.34	0.50
	1576-96-1	Anti-2-pentol	769	888	C_5_H_12_O	2.26	2.26	2.45	2.65	3.17
	6032-29-7	2-amyl alcohol	691	795	C_5_H_10_O	0.77	0.68	0.74	0.86	1.24
Ketone	23726-93-4	Damascus ketone	1386	1496	C_13_H_18_O	0.11	0.05	0.06	0.05	0.08
	110-43-0	2-Geng ketone	891	984	C_7_H_14_O	7.98	7.31	7.13	6.94	6.14
	600-14-6	2,3-pentadione	698	788	C_5_H_8_O_2_	6.09	6.26	5.91	6.34	5.94
Pyrrolidine	85213-22-5	2-acetyl-1-pyrrolidine	932	1032	C_6_H_9_NO	0.10	0.04	0.06	0.06	0.15
Aromatic hydrocarbon	95-63-6	1,2,4-trimethylbenzene	993	1039	C_9_H_12_	13.24	14.51	15.58	14.15	11.23
Terpenes	586-62-9	terpinolene	1088	1112	C_10_H_16_	0.18	0.25	0.28	0.23	0.16
Ester	123-92-2	isoamyl acetate	878	945	C_7_H_14_O_2_	0.56	0.66	0.63	0.61	0.58
	623-70-1	ethyl crotonate	835	923	C_6_H_10_O_2_	0.10	0.10	0.11	0.12	0.14
Acid	64-19-7	acetic	619	773	CH_3_COOH	0.06	0.18	0.18	0.23	0.41
Furan	3208-16-0	2-Ethyl furan	703	735	C_6_H_8_O	4.03	6.20	6.16	5.34	4.62

**Table 3 foods-12-03435-t003:** Analysis of the color difference characteristics of PBMAs and beef.

Color Attribute	HP Content (%)					Monascus Red	Beef
	0	1	3	5	7		
Extrudates/Cook thoroughly							
*L**	51.50 ± 0.44 ^b^	51.42 ± 0.79 ^b^	45.84 ± 0.20 ^c^	39.48 ± 1.76 ^d^	36.34 ± 2.41 ^e^	58.31 ± 1.40 ^a^	57.15 ± 1.23 ^a^
*a**	4.67 ± 0.04 ^d^	8.35 ± 0.45 ^c^	10.07 ± 0.79 ^b,c^	11.60 ± 1.02 ^b^	19.25 ± 2.14 ^a^	5.16 ± 0.11 ^d^	9.20 ± 0.48 ^c^
*b**	16.70 ± 0.59 ^a,b^	15.42 ± 0.33 ^a,b^	14.59 ± 1.76 ^b,c^	12.89 ± 1.50 ^c,d^	15.48 ± 1.60 ^a,b^	17.32 ± 0.09 ^a^	11.73 ± 0.28 ^d^
ΔE	47.01 ± 0.33 ^d^	46.47 ± 0.74 ^d^	53.80 ± 0.54 ^c^	57.77 ± 2.18 ^b^	62.02 ± 1.64 ^a^	40.30 ± 1.32 ^e^	41.16 ± 1.18 ^e^
Raw material							
*L**	88.40 ± 3.29 ^a^	75.48 ± 2.30 ^b^	73.44 ± 1.99 ^b,c^	65.74 ± 4.67 ^d^	68.62 ± 0.36 ^c,d^	87.57 ± 1.46 ^a^	38.06 ± 0.68 ^e^
*a**	4.60 ± 0.18 ^f^	5.33 ± 0.16 ^d^	5.17 ± 0.13 ^d^	5.80 ± 0.15 ^c^	6.63 ± 0.02 ^b^	4.88 ± 0.09 ^e^	15.64 ± 0.14 ^a^
*b**	19.16 ± 0.24 ^a^	16.12 ± 0.28 ^b^	13.15 ± 0.44 ^c^	11.87 ± 1.09 ^d^	12.55 ± 0.29 ^c,d^	19.70 ± 0.36 ^a^	10.20 ± 0.12 ^e^
ΔE	21.09 ± 0.89 ^e^	26.14 ± 1.63 ^d^	27.81 ± 2.44 ^c,d^	32.93 ± 3.99 ^b^	30.63 ± 0.27 ^b,c^	20.88 ± 0.34 ^e^	60.43 ± 0.62 ^a^

Note: Different superscript letters in the same columns indicate significant differences (*p* < 0.05).

**Table 4 foods-12-03435-t004:** The total score of sensory evaluation.

Sensory Properties	HP Content (%)				
	0	1	3	5	7
A group					
color	4.71 ± 2.36 ^a,b^	6.86 ± 2.12 ^a^	6.71 ± 2.06 ^a^	4.43 ± 2.51 ^a,b^	3.43 ± 2.57 ^b^
odor	2.57 ± 1.13 ^b^	4.29 ± 2.06 ^a,b^	3.86 ± 2.41 ^a,b^	4.57 ± 1.81 ^a,b^	6.14 ± 2.97 ^a^
E group					
color	2.71 ± 1.38 ^b^	5.71 ± 3.04 ^a^	5.43 ± 2.07 ^a,b^	3.57 ± 1.81 ^a,b^	3.29 ± 1.98 ^a,b^
odor	2.29 ± 1.11 ^b^	4.14 ± 1.46 ^a,b^	3.86 ± 1.68 ^a,b^	4.86 ± 2.79 ^a,b^	5.29 ± 3.09 ^a^

Note: Different superscript letters in the same columns indicate significant differences (*p* < 0.05).

## Data Availability

The datasets generated for this study are available on request to the corresponding author.

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
