# Peer review of "Effects of *Haematococcus pluvialis* Addition on the Sensory Properties of Plant-Based Meat Analogues"

_foods, 2023, doi:10.3390/foods12183435_

Round 1
Reviewer 1 Report
Report on the manuscript foods-2549682 entitled: Effects of Haematococcus pluvialis on flesh-sensitive properties of high moisture meat analogues.
- The research is quite interesting, but I cannot understand the need to use animal-related wording. Why “flesh-sensitive” and/or “meat analogues”?
In fact, the authors used animal terminology in the title and L. 8-11 but nowhere else. Please, delete or rewrite such lines and the title!
Results description and Discussion MUST be improved:
- Table 1. Units?
- Figures 3 and 4 and Table 2 must be discussed together. They MUST be connected or related so please, carry out a deeper analysis of the results and a better description and discussion.
- Figures 5 and 6 and Table 1 must be connected or related so please, carry out a deeper analysis of the results and a better description and discussion.
Moderate editing of English language required.
Some examples:
- L. 12, 23, 257, 311. “…compared to…”.
- L. 218 and 221. “amul”, meaning?
Author Response
Response to Reviewer 1 Comments
Dear reviewer,
Thank you for giving us a chance to revise the manuscript, we have modified the manuscript accordingly, and the specific modification are listed point by point below.
Point 1: The research is quite interesting, but I cannot understand the need to use animal-related wording. Why “flesh-sensitive” and/or “meat analogues”?
Response 1: The word was change in the title of the manuscript. The title has been revised to “Effects of Haematococcus pluvialis addition on the sensory properties of plant-based meat analogues”.
Point 2: Table 1. Units?
Response 2: The units was checked and revised.
Point 3: Figures 3 and 4 and Table 2 must be discussed together. They MUST be connected or related so please, carry out a deeper analysis of the results and a better description and discussion.
Response 3: The result and discussed of Figure 3 and 4 and Table 2 have been revised in the manuscript.
Point 4: Figures 5 and 6 and Table 1 must be connected or related so please, carry out a deeper analysis of the results and a better description and discussion.
Response 4: The discussion of Figure 5 and 6 and Table1 has been revised in the manuscript, and relative sensory evaluation was added in section 3.7.
Point 5: Moderate editing of English language required. Some examples:
- L. 12, 23, 257, 311. “…compared to…”.
- L. 218 and 221. “amul”, meaning?
Response 5: Thank you very much for your advise, the English language of this paper has been revision by MDPI English editing.
Thank you very much for your work concerning my paper.
Wish you all the best!
Sincerely yours,
Meng Liu

Reviewer 2 Report
Reviewer comments
· Line 35, page 1: DHA, should be in full form.
· Line 36, page 1: “But microalgae….” Change the sentence, it shout not start with but.
· Line 37, page 1: “Directly adding is considered a new ..” reconstruct the sentence.
· Line 47, page 2: “…so that restrict its use”, reconstruct the section of sentence.
· Line 61, page 2: “…Nature pigments replace with natural pigments
· Line 64, page 2: “impossible foods”, write as “Impossible Foods”.
· Line 70, page 2: “By Comparing with the green microalgae..”, dot start with by.
· Line 102, page 3: 840 r/min, replace with RPM.
· Line 108, PAGE 3: “2 cm*2 cm*1 cm, replace with 2 cm x 2 cm x 1 cm. Change accordingly in all the similar sections of the text.
· Line 109, page 3: Wobb's cutter and EZ-TEST biomechanical tester, mention the name of manufacturer.
· Line 147, page 4: L*, a*, b*, write in italics.
· Line 192, page 5: Different letters in the same characteristics….., replace with Different superscript letters in the same characteristics…..

Its moderate, but minor corrections can improve that.
Author Response
Response to Reviewer 2 Comments
Dear reviewer,
Thank you for giving us a chance to revise the manuscript, we have modified the manuscript accordingly, and the specific modification are listed point by point below.
Point 1: Line 35, page 1: DHA, should be in full form.
Response 1: The full form (Docosahexaenoic acid) was added in this section.
Point 2: Line 36, page 1: “But microalgae….” Change the sentence, it shout not start with but.
Response 2: The senctence has been revised in the manuscript.
Point 3: Line 37, page 1: “Directly adding is considered a new ..” reconstruct the sentence.
Response 3: The sentence has been revised in the Line 59, page 2.
Point 4: Line 47, page 2: “…so that restrict its use”, reconstruct the section of sentence.
Response 4: The sentence has been revised in the Line 64, page 2.
Point 5: Line 61, page 2: “…Nature pigments replace with natural pigments
Response 5: The Natural pigments was replaced with natural pigments.
Point 6: Line 64, page 2: “impossible foods”, write as “Impossible Foods”.
Response 6: The mistake has been revised.
Point 7: Line 70, page 2: “By Comparing with the green microalgae..”, dot start with by.
Response 7: The sentence has been revised in Line 69, page 2.
Point 8: Line 102, page 3: 840 r/min, replace with RPM.
Response 8: The mistake has been revised.
Point 9: Line 108, PAGE 3: “2 cm*2 cm*1 cm, replace with 2 cm x 2 cm x 1 cm. Change accordingly in all the similar sections of the text.
Response 9: The symbol has been revised in the paper.
Point 10: Line 109, page 3: Wobb's cutter and EZ-TEST biomechanical tester, mention the name of manufacturer.
Response 10: The name of manufacturer has been revised.
Point 11: Line 147, page 4: L*, a*, b*, write in italics.
Response 11: According to your suggestion, the mistake has been revised.
Point 12: Line 192, page 5: Different letters in the same characteristics….., replace with Different superscript letters in the same characteristics…..
Response 12: The sentence was replaced in Line 200, page 5.
Thank you very much for your work concerning my paper and covering these shortcoming.
Wish you all the best!
Sincerely yours,
Meng Liu

Reviewer 3 Report
The manuscript foods-2549682 entitled " Effects of Haematococcus pluvialis on flesh-sensitive properties of high moisture meat analogues”
Overall, the paper is well written and very well explained. I have few observations that need to be addressed to improve the paper’s quality.
In its current state, the level of English throughout your manuscript does not meet the journal's desired standard. There are many badly worded/constructed sentences. Extensive rephasing and paraphrasing led to a change in the actual meaning of the sentences. Difficult to understand.
Introduction: Please rearrange introduction, starting from meat analogues its importance, nutritional benefits, then discuss it issues needed to be addressed, following the solution, using Haematococcus pluvialis.
Materials and Methods
Why the authors did not perform sensory analysis. It is very important indicator in such type of studies. Please include the data if available or possible to conduct sensory analysis on the stored samples.
Results and discussion
Table 1 & 3. Please provide p-value of all parameters or indicate it using stars for better understanding of the level of significance.
Figure 4: Please increase size or split these figures. In the current form it is difficult to understand. Additionally, provide error bars.
Figure 6: Please provide p-value of all parameters or indicate it stars to understand the level of significance.
3.5. Rheological analysis: Why not measured storage modulus (G′) and loss modulus (G′′)?
Conclusion: The conclusion section is redundant with results already summarized and analyzed. Rewrite please.
In its current state, the level of English throughout your manuscript does not meet the journal's desired standard. There are many badly worded/constructed sentences. Extensive rephasing and paraphrasing led to a change in the actual meaning of the sentences. Difficult to understand.
Author Response
Response to Reviewer 3 Comments
Dear reviewer,
Thank you for giving us a chance to revise the manuscript, we have modified the manuscript accordingly, and the specific modification are listed point by point below.
Point 1: In its current state, the level of English throughout your manuscript does not meet the journal's desired standard. There are many badly worded/constructed sentences. Extensive rephasing and paraphrasing led to a change in the actual meaning of the sentences. Difficult to understand.
Response 1: Thank you for your correction and criticism. The paper was English edited[1] by MDPI service after we firstly simple revise.
Point 2: Introduction: Please rearrange introduction, starting from meat analogues its importance, nutritional benefits, then discuss it issues needed to be addressed, following the solution, using Haematococcus pluvialis.
Response 2: I would appreciate for your valuable comments on the introduction section. According to your suggestion, introduction was revise and showed in manuscript.
Point 3: Why the authors did not perform sensory analysis. It is very important indicator in such type of studies. Please include the data if available or possible to conduct sensory analysis on the stored samples.
Response 3: According to your advice, sensory evaluation has been completed and showed in section 2.7 (page 5), and section 3.7 (page 11).
Point 4: Table 1 & 3. Please provide p-value of all parameters or indicate it using stars for better understanding of the level of significance.
Response 4: The note (Different superscript letters in the same columns indicate significant differences (p<0.05).) has been revised showed in Line 243, page 6 and Line 384, page 13.
Point 5: Figure 4: Please increase size or split these figures. In the current form it is difficult to understand. Additionally, provide error bars.
Response 5: The Figure 4 has been revised according to this suggestion. Then Please allow me to explain the reason about the error bars. The Heracles neo result only showed retention index of different volatile compounds, and the retention area as average value of three experiments was obtained after system comparison. We have tried to find original value of different volatile compounds but failed at last. So to verify the repeatability of these result, we put these chromatograms together to comparison. The result showed a excellent repeatability. On the other hand, we can think if the reliability of result is benefited from using data library to analysis.
Point 6: Figure 6: Please provide p-value of all parameters or indicate it stars to understand the level of significance.
Response 6: The Figure 6 has been added p-value and showed in Line 293, page 9.
Point 7: 3.5. Rheological analysis: Why not measured storage modulus (G′) and loss modulus (G′′)?
Response 7: The experiments has been supplemented and showed in Figure 6.
Point 8: Conclusion: The conclusion section is redundant with results already summarized and analyzed. Rewrite please.
Response 8: According to your suggestion, the conclusion has been revised.
Thank you very much for your work concerning my paper and covering these shortcoming.
Wish you all the best!
Sincerely yours,
Meng Liu

Reviewer 4 Report
The manuscript foods-2549682 entitled " Effects of Haematococcus pluvialis on flesh-sensitive properties of high moisture meat analogues”
Abstract
The main content of the abstract should include the briefly purpose of the research, the principal result and major conclusion. The abstract, in the present form is very disorganized. Please revise it with improved English.
Introduction
Authors spend too many spaces in general information about irrelevant topics. Additionally, this section is very disorganized, without any connection between paragraphs. Authors must focus that general information and improve this section. Please rewrite this section
Please incorporate sensory analysis to improve conclusion.
Figure 4: Please improve pixel quality and size.
Conclusion: Conclusion is too generalized, please make it specific and short.
Difficult to understand.
Author Response
Response to Reviewer 4 Comments
Dear reviewer,
Thank you for giving us a chance to revise the manuscript, we have modified the manuscript accordingly, and the specific modification are listed point by point below.
Point 1: The main content of the abstract should include the briefly purpose of the research, the principal result and major conclusion. The abstract, in the present form is very disorganized. Please revise it with improved English.
Response 1: Thank you for your suggestion, the abstract has been revised.
Point 2: Authors spend too many spaces in general information about irrelevant topics. Additionally, this section is very disorganized, without any connection between paragraphs. Authors must focus that general information and improve this section. Please rewrite this section
Response 2: The introduction was composed and revised.
Point 3: Please incorporate sensory analysis to improve conclusion.
Response 3: Sensory evaluation has been added in 3.7 section and conclusion was revised.
Point 4: Figure 4: Please improve pixel quality and size.
Response 4: According to your suggestion, figure 4 has been revised.
Point 5: Conclusion: Conclusion is too generalized, please make it specific and short.
Response 5: Conclusion has been rearranged and written.
Point 6: Quality of English language: difficult to understand.
Response 6: Thank you very much for your criticism and the English language has been edited after simple revise.
Thank you very much for your work concerning my paper and covering these shortcoming.
Wish you all the best!
Sincerely yours,
Meng Liu

Round 2
Reviewer 1 Report
Error bars in Figures are missing.
--
Author Response
Response Reviewer 1 Comments
Dear reviewer,
Thank you very much for your suggestion and encouragement. We have revised and thought carefully. And the specific modification was listed point by point below.
Point 1: Error bars in Figures are missing.
Response 1: According to your suggestion, we have discussed and revised. But please allow me to explain the reason about the Error bars in Figure 4 are missing. The retention index and retention area of different volatile compounds were obtained directly in software after comparison, and the result is the average value of three experiments. Then we tried our best to make sure error value by finding the original data, but false at least. So we put these gas chromatograms together for comparison, and the result showed perfect replication. On the other hand, we can think if the high reliability of the result is beneficial from systematic analysis.
Thank you very much for your work concerning my paper and covering these shortcoming.
Wish you all the best.
Meng Liu

Reviewer 3 Report
Revised manuscript is improved.
Author Response
Response Reviewer 3 Comments
Dear reviewer,
We appreciate your comments and suggestion on the paper. The structure became more compact, the experiments became more complete. The study standard was improved after revision according to your advise. Thank you very much.
Wish you all the best.
Meng Liu

Reviewer 4 Report
I am satisfied with authors response.
Author Response
Response Reviewer 4 Comments
Dear reviewer,
Thank you very much for encouragement and suggestion. According to these advise, the paper was improved and the main idea was emphasized after revision. We appreciate your work for the development of the study.
Wish you all the best.
Meng Liu
